# Novel Computed Tomography Variables for Assessing Tricuspid Valve Morphology: Results from the TRIMA (Tricuspid Regurgitation IMAging) Study

**DOI:** 10.3390/jcm11102825

**Published:** 2022-05-17

**Authors:** Valeria Cammalleri, Myriam Carpenito, Domenico De Stefano, Gian Paolo Ussia, Maria Caterina Bono, Simona Mega, Annunziata Nusca, Nino Cocco, Edoardo Nobile, Aurelio De Filippis, Luka Vitez, Carlo Cosimo Quattrocchi, Francesco Grigioni

**Affiliations:** 1Unit of Cardiovascular Science, Department of Medicine, Campus Bio-Medico University, 00128 Rome, Italy; m.carpenito@policlinicocampus.it (M.C.); m.bono@policlinicocampus.it (M.C.B.); s.mega@policlinicocampus.it (S.M.); a.nusca@policlinicocampus.it (A.N.); n.cocco@policlinicocampus.it (N.C.); e.nobile@unicampus.it (E.N.); a.defilippis@unicampus.it (A.D.F.); f.grigioni@policlinicocampus.it (F.G.); 2Unit of Diagnostic Imaging and Interventional Radiology, Campus Bio-Medico University, 00128 Rome, Italy; d.destefano@policlinicocampus.it; 3Unit of Interventional Cardiology, Department of Medicine, Campus Bio-Medico University, 00128 Rome, Italy; g.ussia@policlinicocampus.it (G.P.U.); c.quattrocchi@policlinicocampus.it (C.C.Q.); 4Department of Cardiology, University Medical Center Ljubljana, 1000 Ljubljana, Slovenia; luka.vitez@kclj.si

**Keywords:** tricuspid valve, transcatheter tricuspid valve intervention, commissures, computed tomography

## Abstract

Background: Computed tomography (CT) is the recommended imaging technique for defining the anatomical suitability for current transcatheter technologies and planning tricuspid valve (TV) intervention. The aim of the Tricuspid Regurgitation IMAging (TRIMA) study was to assess the geometrical characteristics of the TV complex using novel CT parameters. Methods: This prospective, single-center study enrolled 22 consecutive patients with severe tricuspid regurgitation, who underwent a cardiac CT study dedicated to the right chambers. The following variables were obtained: annulus area and perimeter, septal-lateral and antero-posterior diameters, tenting height, and anatomical regurgitant orifice area. Moreover, the following novel annular parameters were assessed: distance between commissures, distance between TV centroid and commissures, and angles between centroid and commissures. Results: A significant phasic variability during the cardiac cycle existed for all variables except for eccentricity, angles, and distance between the postero-septal and antero-posterior commissure and distance between the centroid and antero-posterior commissure. There was a significant relationship between the TV annulus area and novel annular parameters, except for annular angles. Additionally, novel annular variables were found to predict the annulus area. Conclusions: These novel additional variables may provide an initial platform from which the complexity of the TV annular morphology can continue to be better understood for further improving transcatheter therapies.

## 1. Introduction

Anatomic knowledge of the tricuspid valve (TV) is the first step in the diagnostic algorithm of patients with tricuspid regurgitation (TR) who are candidates for transcatheter tricuspid valve intervention (TTVI) [1,2,3,4,5,6,7]. Whereas anatomical information has been historically derived from cadaveric specimens or openly, non-invasive imaging techniques now provide a complete understanding of the anatomy of the cardiac structures. Due to the three-dimensional (3D) elliptical configuration of the TV apparatus and its position in the chest, detailed morphological visualization of the cardiac structures can be a challenge using echocardiography. However, tailored 3D echocardiography software that accounts for the nonplanarity of the TV annulus is potentially a very reliable method of assessing TV geometry and dynamics because of the high temporal resolution of echocardiography [8]. Nevertheless, the anatomic changes of TV and the remodeling of the right chambers in patients with TR are best assessed with computed tomography (CT) imaging, owing to the acquisition of 3D data with a high spatial resolution [1,2,3,4,5,6,7,8,9,10,11,12]. CT imaging can shed light on the underlying mechanism of TR, defining the anatomical suitability for current transcatheter technologies, and planning TV intervention. 

Each TTVI needs specific imaging protocols to evaluate the anatomical feasibility and outcomes of the potential procedure. Several parameters have been studied so far and they have been found to help in the decision-making process and support the development of novel transcatheter therapies [9,10,11,12,13,14,15].

The purpose of the present study was to assess the geometrical characteristics of the TV complex using novel CT scan parameters and to explore relationships between these anatomical findings.

## 2. Materials and Methods

### 2.1. Patient Population

This prospective, single-center study enrolled 22 consecutive patients with symptomatic TR referred to University Hospital Campus Bio-Medico of Rome for TTVI consideration, between June 2020 and October 2021. Patients were included in the study only if they presented with TR that was at least severe, diagnosed by echocardiography, and underwent a cardiac CT study dedicated to the right chambers. Exclusion criteria were as follows: previous TV surgery, rheumatic TV disease, active endocarditis, congenital heart disease, and need of urgent cardiac surgery. Tricuspid regurgitation was assessed according to semi-quantitative echocardiographic color flow doppler evaluation [16,17]. CT data were prospectively collected in the institutional medical imaging archiving server and retrospectively analyzed. The study protocol was approved by the local ethical committee with the name TRIMA (Tricuspid Regurgitation IMAging) study. All patients provided written consent for taking part to this study.

### 2.2. CT Acquisition

CT examinations were performed using a 128-slice multidetector CT scanner (Siemens Somatom Definition AS+, Siemens Erlangen, Erlangen, Germany) with a collimation of 128 × 0.6 mm. ECG-assisted data acquisition was performed with a retrospective ECG gating without dose modulation, in order to cover the entire cardiac cycle (R-R) at the same dosage. Tube voltage varied for each exam from 100 to 120 kV, depending on the patient’s body mass index (BMI) and glomerular filtration rate (GFR). Based on the patients’ BMI and GFR, ionic contrast medium (Omnipaque 350 mg I/mL, GE Healthcare, Chicago, IL, USA) was infused in the antecubital vein with a biphasic contrast protocol: 60–80 mL mixture of 80%/20% contrast/saline, with a flow rate of 4.0–5.0 mL/s, followed by a 50 mL of saline. Initiation of scanning was synchronized to the arrival of the contrast material in the main pulmonary trunk, using automated peak enhancement detection with a threshold of 120 Hounsfield Units (HU). The scan target window included the entire heart, from the superior vena cava to the supra hepatic inferior vena cava level. Thus, CT datasets were reconstructed as an axial thin-sliced image (0.6–0.75 mm, with soft-tissue convolution kernel and iterative reconstruction algorithm, depending on the level of image noise) at each 5% of the R-R interval, covering the entire cardiac cycle with a multi-phase set. ECG editing was used in case of uncorrected sync.

### 2.3. CT Data Analysis

For offline image analysis, the reconstructions were transferred to an external workstation (3mensio Structural Heart; Pie Medical Imaging, Maastricht, The Netherlands).

Measurements of the TV annulus, named area and perimeter, were assessed using the 3mensio semi-automated software. The septal-lateral and the antero-posterior annular diameters were manually obtained. The septal-lateral diameter was measured as the maximal distance in the septal to lateral direction and coincides with the annulus measurement in the four-chamber view; the antero-posterior diameter is orthogonal to the septal-lateral one and coincides with the measurement in the two-chamber view [9]. The eccentricity was calculated as antero-posterior/septal-lateral diameter [9]. Then, the automatic identification of the commissures was reviewed, and the commissures were located from their unique anatomical findings and folded configuration, which identified them as ‘commissural leaflets’ [18,19]. They were named antero-septal (AS), postero-septal (PS), and antero-posterior (AP) commissures. The length of each ‘commissural leaflet’ was measured in the end-diastole. Furthermore, starting from the commissures, we calculated and tested novel annular measurements. Specifically, in the short-axis view, the distance between the commissures was measured, resulting in a triangle, with the vertices at the level of the commissures. The geometrical centroid (Ce) of the triangle was then identified in the short-axis view, and the distance between the centroid and commissures was recorded, as well as the angles formed by the centroid and two contiguous commissures, named alpha (AS-Ce-PS), beta (PS-Ce-AP), and gamma (AP-Ce-AS), respectively (Figure 1).

All annular measurements were taken in both end-systole and end-diastole. Using the multiplanar reconstruction, we obtained right atrium (RA) and right ventricle (RV) dimensions in two long-axis orthogonal views (four- and two-chamber views). The distance between the annulus and the atrial roof and the distance between the annulus and the ventricular apex were measured in both end-systole and end-diastole. Finally, multiplanar reconstruction at mid-systole was used to determine the degree of tethering of each leaflet, the leaflets tenting height (or coaptation depth) in four- and two-chamber views, and the anatomical regurgitant orifice area (AROA) [3,9,14,20] (Figure 2). Additional measurements of the anatomical regurgitant orifice (ARO) were obtained, named the perimeter, maximal and minimum distance, and gap between the septal and anterior leaflet (SA-gap) and between the septal and posterior leaflet (SP-gap).

All steps of the analysis are summarized in Appendix A.

### 2.4. Statistical Analysis

Categorical variables are expressed as frequencies and percentages; continuous variables are presented as mean ± standard deviation or medians and interquartile ranges (IQRs), as appropriate. The phasic variability (diastole/systole) of anatomical findings was quantitatively evaluated and compared by a paired-samples *t*-test. Other intragroup comparisons were completed using a paired-samples *t*-test and Wilcoxon signed-rank test for non-parametric variables. The Pearson correlation was used to explore the relationship between measurements. Linear multiple regression was used to test novel CT variables associated with the 3D-automated annulus area. To avoid multi-collinearity, the correlation between the tricuspid annulus dimensions (distance between commissures, and distance between centroid and commissures) and annulus area was tested separately. Results are reported as point estimates and 95% confidence intervals (CI). Differences were considered significant at *p* < 0.05. Statistical analyses were performed using IBM SPSS version 26 (IBM, Armonk, NY, USA).

## 3. Results

The mean age of the overall population was 78 ± 7.66 years and 17 (77.3%) were females. All patients suffered from TR that was equal to or greater than severe. The mechanism was predominantly functional (atrial 68%, ventricular 14%), whereas four patients (18%) had a mixed etiology (lead-induced and functional) [21,22]. Baseline clinical characteristics of our study population are summarized in Table 1. 

A three-leaflet configuration of the TV was observed in all patients; nine (40.9%) presented a multi-scalloped anterior or posterior leaflet with the three commissures always identifiable. Tricuspid annulus measurements and differences between diastole and systole are reported in Table 2.

The annular dimensions were generally observed to reduce from diastole to systole. Specifically, a significant phasic variability existed for all variables except for eccentricity, PS-AP distance, Ce-AP distance, and angles, which remained statistically unvaried during the cardiac cycle. On the other hands, right chamber measurements significantly changed, with the RA increasing in systole and the RV reducing in systole.

The tenting height was significantly greater in two-chamber view (9.45 ± 3.05 mm) when compared with the four-chamber view (8.29 ± 3.49 mm) (difference −1.16; 95%; CI −2.18 to −0.136; *p* = 0.028). The degrees of tethering of the anterior, septal, and posterior leaflets were in median 11.5° (IQR 5–19.5), 14.5° (7.25–20.5), and 10° (4.5–12), respectively.

Functional CT analysis for AROA was feasible in all patients. The median tricuspid AROA and perimeter were 1.11 cm^2^ (IQR 0.90–2.14) and 62.60 mm (49.58–70.40), respectively.

The maximal and minimum distances were 17.95 mm (16.63–23.63) and 8.10 mm (6.28–11.08). The major gap existed between the septal and the anterior leaflet (6.75 mm, IQR 5.90–9.63) compared to the septal and posterior leaflet (5.80 mm, IQR 4.65–6.53, *p* = 0.046). A significant correlation between tethering angles and AROA was observed for the anterior (r = 0.743, *p* < 0.001), septal (r = 0.731, *p* < 0.001), and posterior (r = 0.737, *p* < 0.001) leaflet.

There was no statistically significant difference in the length of ‘commissural leaflets’, although the AP was the longest (6.84 ± 3.06 mm), compared to the AS (6.5 ± 1.49 mm) and PS commissure (6.32 ± 1.99 mm).

Correlations between measurements were reported in Table 3.

Interestingly, there was a significant relationship between TV annulus area and novel annular parameters, in both systole and diastole, except for the annular angles (Figure 3).

No correlation was found between tenting height and annulus area and tenting height and right chambers’ dimensions. Similarly, no correlation existed between the TV annulus area and AROA measurements, except for the SA-gap (r = 0.388, *p* = 0.037).

Applying multiple regression, novel annular variables were found to predict the 3D annulus area. The analysis demonstrated a strong correlation between commissural distance and the TV annulus area in diastole (r^2^ = 0.938) and systole (r^2^ = 0.944); moreover, a strong correlation was found between the centroid commissure distance and TV annulus area in diastole (r^2^ = 0.925) and systole (r^2^ = 0.944). All variables added statistical significance to the prediction (Table 4).

## 4. Discussion

The present study confirms that CT provides a comprehensive assessment of the anatomy and geometry of the TV structure in patients suffering from severe or greater functional TR and candidates to TTVI. Particularly, we observed the following: (1) the TV annular dimension assessed using conventional CT scan parameters agrees with previously reported data in this setting of patients; (2) dilation of the tricuspid annulus preferentially occurs along its free-wall distance, moving away the AP commissure from the TV centroid; and (3) the use of novel variables for assessing TV annular morphology, significantly correlated with conventional measurements of the tricuspid annulus.

### 4.1. Annulus Sizing

The annulus was typically flatter and more circular in patients with functional TR, which diminished the typical saddle shape. Additionally, in the presence of TR, the percent reduction of the tricuspid annulus size during the systole is significantly decrease. Data from a CT study of van Rosendael PJ and co-workers showed that systolic tricuspid annular perimeter and area were significantly larger in patients with severe TR than those with lower regurgitation (145.3 ± 14.4 vs. 129.2 ± 12.8 mm, *p* < 0.001, and 1539.7 ± 260.2 vs. 1228.4 ± 243.5 mm^2^, *p* < 0.001, respectively) [9]. In our study population, including all cases of at least severe TR, the annular area and perimeter, determined by a semi-automated software, were similar to those previously reported. Moreover, the eccentricity index of 0.99 ± 0.12 in systole and 0.99 ± 0.10 in diastole indicates an almost perfectly circular shape of the TV annulus during the entire cardiac cycle, confirming what was already assumed [23].

### 4.2. Novel Annular Parameters 

In the present study, we proposed the use of additional annular measurements, which may provide further information about the morphology of the TV apparatus. These parameters include the following: distance between commissures, distance between TV centroid and commissures, and angles between centroid and commissures. Starting from their peculiar anatomical finding, the identification of commissures is the critical point for assessing these variables. The commissures are supported by fan-shaped chords and do not open directly into the annulus, but a few millimeters of the leaflet tissue remain, similar to small scallops [7,18]. Precisely, the AS commissure is placed anteriorly, just below the first right coronary tract and the anterior aortic valve cusp. At this site, it is possible to observe a short fan-shaped chorda, arising directly from the septal band of the crista supraventricularis or from a small papillary muscle on that band [18]. The PS commissure is placed posteriorly after where the coronary sinus enters the atrium. Anatomically it has three landmarks: a fan-shaped chorda, a papillary muscle, and a fold in the septal leaflet [18]. The AP commissure is placed in correspondence of the free RV wall, usually below the right coronary artery, roughly at the acute margin [18]. It is well identified by a fan-shaped chorda and the anterior papillary muscle, which points towards this commissure. Commissures folded during systole to serve as ends of the zone of apposition between leaflets are not points that can be marked on valve annulus, but typically account for approximately 25% of the annular length of the TV [24]. For this reason, some anatomists have described them as ‘commissural leaflets’ [19,24]. Our experience has shown that as they are easily identifiable and measurable in end-diastole, with the AP resulting as the longest one.

Anatomical variants in the number and location of leaflets exist and have been differently described by pathological and surgical studies, but all these assumptions derive from the embryogenesis of the TV [19,24,25,26]. Embryologically, the TV has a septal leaflet and a mural or free-wall leaflet. As the free-wall leaflet folds and changes shape during the cardiac cycle, it acquires a hinge point, thus forming a cleft within this mural leaflet to form the anterior and posterior leaflets and the typical saddle configuration [25,27]. Conversely, the septal leaflet originates from a delamination muscular ventricular septum together with the posteroinferior endocardial cushion [28]. Thus the three-leaflet configuration usually occurs in 28% to 58% of cases, whereas in the remaining cases, additional leaflets or supernumerary commissures with functional folds of tissue have been reported [24,26,29]. A two-leaflet configuration has rarely been described, where the anterior and posterior leaflets are not clearly separated [26]. Our patient population had typical three-leaflet configuration, with 40.9% cases of multi-scalloped anterior or posterior leaflet. However, the main AS, PS, and AP commissures were always identifiable in these cases. Nevertheless, data from published studies have shown that complex TV anatomy with four-leaflet morphology was associated with worse clinical, pathological, and echocardiographic outcomes after transcatheter edge-to-edge repair [30,31]. This may highlight the need to improve anatomic knowledge of the TV apparatus in candidates for transcatheter edge-to-edge repair. On the other hand, this also highlights the need to support, from an anatomic perspective, the development of new technologies that do not exclusively target the leaflets.

The analysis of phasic variability during the cardiac cycle showed that AS-PS, AP-AS, Ce-AS, and Ce-PS distances significantly reduced in systole, whereas PS-AP, Ce-AP, and angles remained unchanged. PS-AP and Ce-AP were the longest commissural distances and the centroid–commissure distances, respectively. In our opinion, these results reflect the process through which secondary TR and remodeling of the tricuspid annulus occur. In case of severe functional TR, the increase in tricuspid annulus dimensions is more pronounced along its free-wall aspect [9,32]. Consequently, annular dilation is asymmetric, mostly involving the anterior and posterior leaflets; thus, the AP commissure is moved away from centroid of the TV. In addition, the lack of phasic variability we observed suggests the absence of tricuspid annular motion along its longest diameter, which could be an expression of advanced annular remodeling associated with annular or RV dysfunction [33,34].

The lack of significant variability in the commissural angles is a critical point to keep in mind when translating anatomical knowledge into technical procedural issues. This result emphasizes the stability, during the cardiac cycle, of the anatomical relationship between the different structures of the TV apparatus, especially the commissures and the annular centroid. Therefore, this assumption may result in stability of devices implanted into the TV annulus.

Finally, the analysis of novel annular variables showed a significant correlation between commissural distances and TV area and between the centroid–commissure distance and TV annulus. Moreover, we found that these measurements significantly predict the three-dimensional annulus area, both in diastole and systole.

### 4.3. Leaflet Coaptation

Using the multiplanar reconstruction, it is possible to obtain additional planes parallel to the TV for optimizing the visualization of the tricuspid leaflets, the mode of coaptation, and the grade of leaflet tethering [7,8,20]. According to the geometrical differences between atrial and ventricular functional TR, a significant correlation occurred between the degree of tethering of all leaflets and AROA, but no relationship existed between the degree of tethering and annulus area. Moreover, the height of leaflet tethering, expressed as coaptation depth, was assessed both in the four-chamber and two-chamber view, resulting as significantly greater in the two-chamber view. This reflects the anatomical saddle-shape configuration of the TV annulus, which, although flattened, always maintains two higher points at the AS commissure and postero-lateral segment level.

Recently, the AROA assessed by CT scan has been proposed as a potential flow-independent parameter of TR severity, supplementing and enriching the traditional echocardiographic parameters [3,14]. This is the first study to examine novel anatomical variables of the anatomical regurgitant orifice and assess the relationship between ARO measurements and the TV annulus area. Particularly, we did not observe any correlation between the TV annulus area and ARO measurements, except for the gap between the septal and the anterior leaflet. This observation remarks that annular dilatation occurs asymmetrically along the free wall of the TV. The anterior leaflet functionality is known to be strongly affected by RV and TV annular dilation, in that its structural constraints (the anterior portion of the annulus and the anterior papillary muscle) are extremely free to move. This makes the anterior leaflet highly susceptible to be pulled away from the relatively fixed septal leaflet and subject to tethering.

### 4.4. Future Directions

These results, seemingly intuitive in terms of anatomical relationships, have been shown to correlate with parameters currently used for tricuspid annular analysis and for device suitability assessment. In our opinion, they could be placed in the perspective of carrying out a more detailed analysis of the tricuspid valve structure, parallel to the growth and development of new transcatheter therapies. Therefore, they would expand the current anatomical knowledge of the TV apparatus and provide new anatomical concepts for new technologies.

### 4.5. Study Limitation

The main limitation of this study is the single-site data collection and the small sample size, although no other studies have analyzed these novel CT-derived parameters. Due to the small sample size, we cannot draw meaningful statistical conclusions and consider it a preliminary and hypothesis-generating study. In addition, we do not have a control group with which to compare the results obtained, and we have not compared our CT findings with other imaging techniques, such as echocardiography. From a methodological point of view, the new measurements do not take into account the non-planarity of the TV and might underestimate the true 3D dimensions. It is hoped that newer iterations of CT three-dimensional software, together with a further comparison between imaging techniques and different populations, will address this issue in the future. Our future research goal is to refine and expand our dataset, in order to overcome these limitations, and analyze our data in relation to TTVI results. Future studies are warranted to validate these parameters and investigate their association with TTVI procedural success and clinical outcomes.

## 5. Conclusions

The TRIMA study provides novel additional parameters derived from CT acquisition specific for right chambers. These data may be used to create an initial platform from which the complexity of the TV annular morphology can continue to be better understood for further improving transcatheter therapies. Finally, these novel CT anatomical findings may have future clinical implications in the complex decision-making process that starts with patient selection and leads to procedural TTVI success.

## Figures and Tables

**Figure 1 jcm-11-02825-f001:**
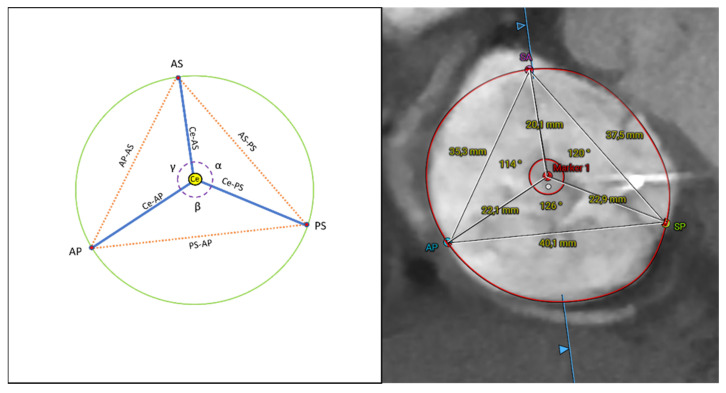
Analysis overview of the novel CT scan measurements. α: alpha; AP: anter-posterior commissure; AS: antero-septal commissure; β: beta; Ce: centroid; γ: gamma PS: postero-septal commissure; SA: septal-anterior commissure; SP: septal-posterior commissure.

**Figure 2 jcm-11-02825-f002:**
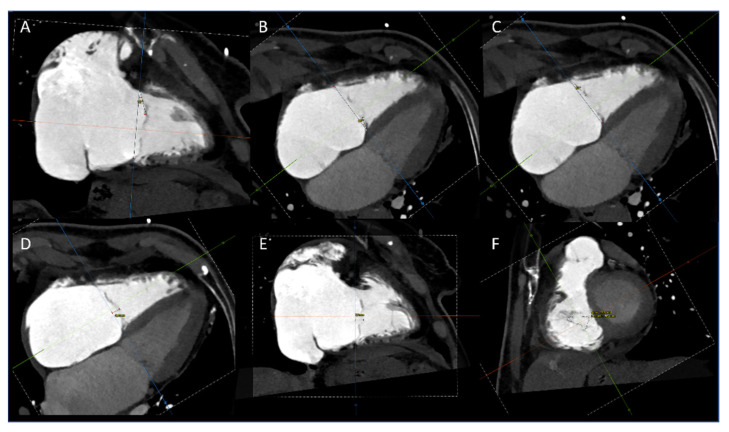
Assessment of leaflets tenting, measuring the degree of tethering of the anterior (**A**), septal (**B**), and posterior (**C**) leaflet; the height of tenting in two-chamber (**D**) and four-chamber (**E**) view; and the anatomical regurgitant orifice area (**F**).

**Figure 3 jcm-11-02825-f003:**
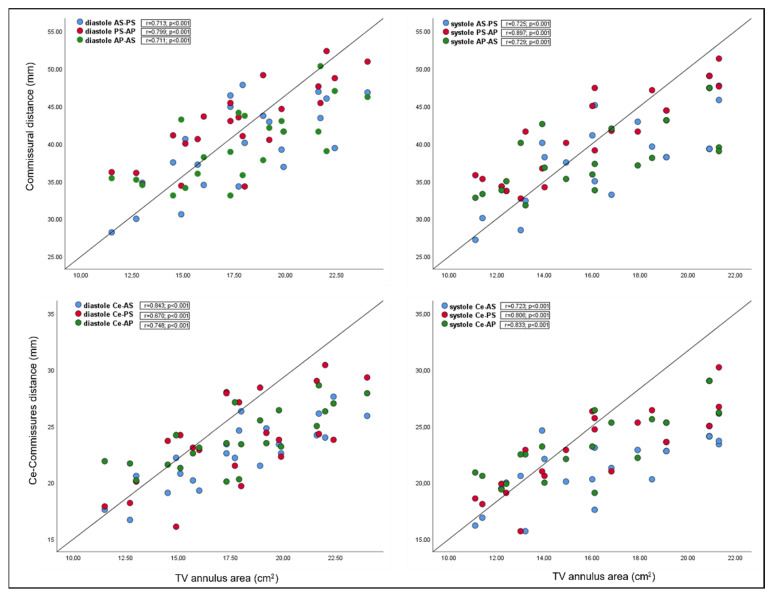
Correlations between the TV annular area and novel annular dimensions (distance between commissures and distance between centroid and commissures) in diastole (left panel) and systole (right panel). AS: antero-septal commissure; AP: antero-posterior commissure; Ce: centroid; PS: postero-septal commissure; TV: tricuspid valve.

**Table 1 jcm-11-02825-t001:** Baseline clinical characteristics.

Patient Characteristics	N = 22
**Mean age**, years old ± SD	78 ± 7.66
**Female**, *n* (%)	17 (77.3)
**BMI**, kg/m^2^ ± SD	24.42 ± 3.89
**Previous CABG**, *n* (%)	2 (9)
**Previous left side valve surgery**, *n* (%)	3 (14)
**Atrial fibrillation**, *n* (%)	20 (91)
**Heart rate**, bpm ± SD	78.36 ± 14.45
**Permanent pacemaker**, *n* (%)	6 (27)
**ICD/CRT**, *n* (%)	4 (18)
**LVEF**, % ± SD	46.27 ± 10.69
**TR grade**, *n* (%)	
3+	17 (77)
4+	2 (9)
5+	3 (14)

BMI: body mass index; CABG: coronary artery bypass graft; ICD/CRT: implantable cardioverter-defibrillator/cardiac resynchronization therapy; LVEF: left ventricular ejection fraction; TR: tricuspid regurgitation.

**Table 2 jcm-11-02825-t002:** Tricuspid annulus measurements and differences between diastole and systole.

Variable	Diastole	Systole	Difference (95% CI)	*p* Value
**Annulus area (cm^2^ ± SD)**	17.78 ± 3.39	16.41 ± 3.47	1.37 (0.74 to 1.99)	<0.001
**Anulus perimeter (mm ± SD)**	152.32 ± 16.32	147.63 ± 15.45	4.69 (1.32 to 8.06)	0.009
**SL diameter (mm ± SD)**	48.27 ± 6.40	45.92 ± 6.26	2.35 (1.30 to 3.40)	<0.001
**AP diameter (mm ± SD)**	47.03 ± 3.63	44.98 ± 3.76	2.05 (1.16 to 2.94)	<0.001
**Eccentricity (index ± SD)**	0.99 ± 0.10	0.99 ± 0.12	−0.01 (−0.03 to 0.02)	0.572
**AS-PS (mm ± SD)**	39.64 ± 5.89	37.58 ± 5.40	2.06 (1.03 to 3.10)	<0.001
**PS-AP (mm ± SD)**	42.48 ± 5.31	41.86 ± 5.93	0.61 (−0.54 to 1.77)	0.280
**AP-AS (mm ± SD)**	39.72 ± 4.88	38.75 ± 4.86	0.97 (0.07 to 1.87)	0.037
**Ce-AS (mm ± SD)**	22.53 ± 2.88	21.19 ± 2.71	1.34 (0.85 to 1.83)	<0.001
**Ce-PS (mm ± SD)**	23.91 ± 3.98	23.06 ± 3.46	0.85 (0.07 to 1.64)	0.034
**Ce-AP (mm ± SD)**	23.84 ± 2.57	23.76 ± 3.15	0.07 (−0.67 to 0.82)	0.841
**α (° ± SD)**	116.32 ± 10.05	115.82 ± 9.25	0.50 (−1.92 to 2.92)	0.672
**β (° ± SD)**	125.50 ± 7.60	126.00 ± 6.60	−0.50 (−3.00 to 2.00)	0.682
**γ (° ± SD)**	117.68 ± 9.78	118.18 ± 8.46	−0.5 (−2.96 to 1.96)	0.677
**RA 4 chambers (mm ± SD)**	65.57 ± 11.57	70.79 ± 9.34	−5.22 (−8.29 to 2.15)	0.002
**RA 2 chambers (mm ± SD)**	63.34 ± 12.32	67.85 ± 10.01	−4.52 (−8.54 to −0.50)	0.030
**RV 4 chambers (mm ± SD)**	70.07 ± 7.85	58.80 ± 8.00	11.27 (7.90 to 14.64)	<0.001
**RV 2 chambers (mm ± SD)**	67.95 ± 7.49	57.57 ± 8.26	10.38 (6.79 to 13.96)	<0.001

CI: confidence interval; SD: standard deviation; SL: septo-lateral; AP: antero-posterior; AS: antero-septal; PS: postero-septal; Ce: centroid; RA: right atrium; RV: right ventricle.

**Table 3 jcm-11-02825-t003:** Pearson correlation between CT variables and annulus area.

Variables	Diastole	Systole
**Annulus perimeter**	r = 0.92 *p* = <0.001	r = 0.98 *p* = <0.001
**SL diameter**	r = 0.93 *p* = <0.001	r = 0.92 *p* = <0.001
**AP diameter**	r = 0.87 *p* = <0.001	r = 0.86 *p* = <0.001
**Eccentricity index**	r = −0.61 *p* = 0.001	r = −0.52 *p* = 0.001
**AS-PS**	r = 0.72 *p* = <0.001	r = 0.72 *p* = <0.001
**PS-AP**	r = 0.80 *p* = <0.001	r = 0.90 *p* = <0.001
**AP-AS**	r = 0.71 *p* = <0.001	r = 0.73 *p* = <0.001
**Ce-AS**	r = 0.84 *p* = <0.001	r = 0.72 *p* = <0.001
**Ce-PS**	r = 0.67 *p* = <0.001	r = 0.81 *p* = <0.001
**Ce-AP**	r = 0.75 *p* = <0.001	r = 0.83 *p* = <0.001
**α**	r = 0.07 *p* = 0.384	r = −0.01 *p* = 0.495
**β**	r = 0.17 *p* = 0.230	r = 0.34 *p* = 0.062
**γ**	r = −0.19 *p* = 0.196	r = −0.26 *p* = 0.121
**RA 4Ch**	r = 0.53 *p* = 0.005	r = 0.56 *p* = 0.003
**RA 2Ch**	r = 0.51 *p* = 0.008	r = 0.24 *p* = 0.138
**RV 4Ch**	r = 0.50 *p* = 0.008	r = 0.34 *p* = 0.059
**RV 2Ch**	r = 0.53 *p* = 0.005	r = 0.48 *p* = 0.011
**Tenting 4Ch**	-	r = 0.01 *p* = 0.492
**Tenting 2Ch**	-	r = 0.35 *p* = 0.057
**Anterior tethering degree**	-	r = 0.24 *p* = 0.301
**Septal tethering degree**	-	r = 0.98 *p* = 0.682
**Posterior tethering degree**	-	r = 0.39 *p* = 0.084
**EROA Area**	-	r = 0.26 *p* = 0.117
**EROA Perimeter**	-	r = 0.26 *p* = 0.12
**EROA Max**	-	r = 0.36 *p* = 0.051
**EROA Min**	-	r = 0.28 *p* = 0.106
**EROA SA-gap**	-	r = 0.39 *p* = 0.037
**EROA SP-gap**	-	r = 0.22 *p* = 0.167

SL: septo-lateral; AP: antero-posterior; AS: antero-septal; PS: postero-septal; Ce: centroid; RA: right atrium; Ch: chamber; RV: right ventricle; EROA: EROA SA-gap: gap between the septal and anterior leaflet; EROA: anatomical regurgitant orifice area; SA-gap: septal anterior leaflets gap; SP-gap: septal posterior leaflets gap.

**Table 4 jcm-11-02825-t004:** Linear multiple regression analysis of novel CT variables associated with automated annulus area: estimated model coefficients and statistical significance of the independent variables.

	Diastole			Systole		
Variables	Coefficient	95% CI	*p* Value	Coefficient	95% CI	*p* Value
**AS-PS**	0.228	0.133–0.322	<0.001	0.205	0.111–0.300	<0.001
**PS-AP**	0.227	0.117–0.338	<0.001	0.297	0.197–0.397	<0.001
**AP-AS**	0.364	0.273–0.456	<0.001	0.252	0.150–0.354	<0.001
**Ce-AS**	0.541	0.332–0.749	<0.001	0.411	0.234–0.588	<0.001
**Ce-PS**	0.310	0.182–0.439	<0.001	0.408	0.262–0.555	<0.001
**Ce-AP**	0.521	0.305–0.737	<0.001	0.508	0.347–0.670	<0.001

CI: confidence interval; AS: antero-septal; PS: postero-septal; AP: antero-posterior; Ce: centroid.

## Data Availability

Data are available using the following link (https://www.dropbox.com/scl/fi/yzas90xd7gn7xg65kbje9/Cammalleri-et-al.JCM-DB.xlsx?dl=0&rlkey=vynwh07wgehfsrt767mnoa6mc (accessed on 6 March 2022).

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
