# Peer review of "Novel Computed Tomography Variables for Assessing Tricuspid Valve Morphology: Results from the TRIMA (Tricuspid Regurgitation IMAging) Study"

_jcm, 2022, doi:10.3390/jcm11102825_

Round 1

Reviewer 1 Report

The authors have analyzed the CT scans of 22 patients to assess the utility of novel annular parameters, namely the distance between the commissures, distance between TV centroid and commissure, and the angles between the centroid and commissures.  They found a significant phasic variability during the cardiac cycle and a significant relationship between distance between the commissures and distance between the centroid and commissures (without a relationship with angles). 

Comments:

  1. This result seems however intuitive, since as the annulus increases in size, the distance between the commissures should necessarily increase (and the distance between the center of the annulus and the edges of the annulus would obviously increase as well). It is difficult to understand how these novel parameters would impact the current analysis of annular area/perimeter and coaptation gaps and heights, for the assessment of device suitability.  Perhaps a discussion of how the authors envision the use of these parameters would be of use.
  2. Could you add a table with the baseline characteristics of the 22 patients included in this study. Were any in atrial fibrillation?  Patients presenting for transcatheter device therapy typically (>70% of cases) are in atrial fibrillation or may have pre-existing pacemakers.  Please comment.
  3. Please discuss how your findings related add to the study by Addetia K, Muraru D, Veronesi F, Jenei C, Cavalli G, Besser SA, Mor-Avi V, Lang RM, Badano LP. 3-Dimensional Echocardiographic Analysis of the Tricuspid Annulus Provides New Insights Into Tricuspid Valve Geometry and Dynamics. JACC Cardiovasc Imaging. 2019 Mar;12(3):401-412. doi: 10.1016/j.jcmg.2017.08.022. Epub 2017 Nov 15. PMID: 29153573. In this study, greater “temporal resolution” allowed for an analysis of annular dynamism.
  4. Your acquisition protocol and scanner technology raises questions as to the resolution of the cardiac structures.  Please see the following (and comment): Pulerwitz TC, Khalique OK, Leb J, Hahn RT, Nazif TM, Leon MB, George I, Vahl TP, D'Souza B, Bapat VN, Dumeer S, Kodali SK, Einstein AJ. Optimizing Cardiac CT Protocols for Comprehensive Acquisition Prior to Percutaneous MV and TV Repair/Replacement. JACC Cardiovasc Imaging. 2020 Mar;13(3):836-850. doi: 10.1016/j.jcmg.2019.01.041. Epub 2019 Aug 14. PMID: 31422136.
  5. Given the differences between atrial functional and ventricular functional tricuspid regurgitation (See: Muraru D et al. Eur Heart J Cardiovasc Imaging. 2021 May 10;22(6):660-669) the tethering angles should have correlated with at least the AROA. Did you evaluate this?
  6. You have confirmed the non-planar nature of the tricuspid annulus in showing the difference in tenting heights between orthogonal views: how does non-planarity affect the novel measurements?
  7. Although the main commissures AS, PS and AP are always identifiable on these 22 patients, there is clearly both pathologic, surgical and echocardiographic evidence for the additional complexity of the valve which significant impacts outcomes and cannot be underestimated. See: 1) Sugiura A, Tanaka T, Kavsur R, Öztürk C, Vogelhuber J, Wilde N, Becher MU, Zimmer S, Nickenig G, Weber M. Leaflet Configuration and Residual Tricuspid Regurgitation After Transcatheter Edge-to-Edge Tricuspid Repair. JACC Cardiovasc Interv. 2021 Oct 25;14(20):2260-2270. doi: 10.1016/j.jcin.2021.07.048. Epub 2021 Aug 23. PMID: 34424200, and 2) Kitamura M, Kresoja KP, Besler C, Leontyev S, Kiefer P, Rommel KP, Otto W, Forner AF, Ender J, Holzhey DM, Abdel-Wahab M, Thiele H, Borger MA, Hahn RT, Lurz P, Noack T. Impact of Tricuspid Valve Morphology on Clinical Outcomes After Transcatheter Edge-to-Edge Repair. JACC Cardiovasc Interv. 2021 Jul 26;14(14):1616-1618. doi: 10.1016/j.jcin.2021.03.052. Epub 2021 May 26. PMID: 34052157.  Your current discussion does not accurately reflect the importance of these complex morphologies.
  8. The study by van Rosendael (van Rosendael PJ, Joyce E, Katsanos S, Debonnaire P, Kamperidis V, van der Kley F, Schalij MJ, Bax JJ, Ajmone Marsan N, Delgado V. Tricuspid valve remodelling in functional tricuspid regurgitation: multidetector row computed tomography insights. Eur Heart J Cardiovasc Imaging. 2016 Jan;17(1):96-105. doi: 10.1093/ehjci/jev140. Epub 2015 Jun 9. PMID: 26060205) showed no difference in eccentricity index with severe vs non-severe TR however the AP dimension, was highly predictive of TR severity. Were you able to correlate any of your measures with TR severity?

Author Response

  1. This result seems however intuitive, since as the annulus increases in size, the distance between the commissures should necessarily increase (and the distance between the center of the annulus and the edges of the annulus would obviously increase as well). It is difficult to understand how these novel parameters would impact the current analysis of annular area/perimeter and coaptation gaps and heights, for the assessment of device suitability. Perhaps a discussion of how the authors envision the use of these parameters would be of use.

Special thanks to you for your valuable comment. A new paragraph entitledFuture directions” has been added at the end of the discussion to address your issue approximatively. Briefly, we think that these new anatomical conceptions could influence and support the development of new technologies.

  1. Could you add a table with the baseline characteristics of the 22 patients included in this study. Were any in atrial fibrillation? Patients presenting for transcatheter device therapy typically (>70% of cases) are in atrial fibrillation or may have pre-existing pacemakers.  Please comment.

According to your suggestion, a table including clinical baseline characteristics of the 22 patients has been added

  1. Please discuss how your findings related add to the study by Addetia K, Muraru D, Veronesi F, Jenei C, Cavalli G, Besser SA, Mor-Avi V, Lang RM, Badano LP. 3-Dimensional Echocardiographic Analysis of the Tricuspid Annulus Provides New Insights Into Tricuspid Valve Geometry and Dynamics. JACC Cardiovasc Imaging. 2019 Mar;12(3):401-412. doi: 10.1016/j.jcmg.2017.08.022. Epub 2017 Nov 15. PMID: 29153573. In this study, greater “temporal resolution” allowed for an analysis of annular dynamism.

Thank you for this essential comment. Our study only examines CT -scan parameters and no correlation was made with echo has been made (Please see Study Limitation section). However, your comment points to valuable concepts about TV morphology and dynamics, that were reported and discussed in the paper.

  1. Your acquisition protocol and scanner technology raises questions as to the resolution of the cardiac structures. Please see the following (and comment): Pulerwitz TC, Khalique OK, Leb J, Hahn RT, Nazif TM, Leon MB, George I, Vahl TP, D'Souza B, Bapat VN, Dumeer S, Kodali SK, Einstein AJ. Optimizing Cardiac CT Protocols for Comprehensive Acquisition Prior to Percutaneous MV and TV Repair/Replacement. JACC Cardiovasc Imaging. 2020 Mar;13(3):836-850. doi: 10.1016/j.jcmg.2019.01.041. Epub 2019 Aug 14. PMID: 31422136.

Thank you for your comments, but the protocol reflects daily clinical practice at our Institution. The protocol was finalized and optimized as reported after trying other acquisition methods (not included in the study). For instance, Figure S1 and Figure 2 show the result we have. The main challenge is the visualization of the commissural leaflets, but a multi-step method including different anatomical landmarks helps in this task.

  1. Given the differences between atrial functional and ventricular functional tricuspid regurgitation (See: Muraru D et al. Eur Heart J Cardiovasc Imaging. 2021 May 10;22(6):660-669) the tethering angles should have correlated with at least the AROA. Did you evaluate this?

Thank you so much for this essential comment. We reported more detailed data on the mechanism of TR in our study population. Initially, we did not report data that correlated with the severity of TR because our population included only patients with at least severe regurgitation. However, according to your suggestion, we calculated the tethering degree for each leaflet and performed a Pearson correlation. Interestingly, there was a significant relationship between the grade of tethering and AROA. These results were included and discussed. Figure 2 was added.  

  1. You have confirmed the non-planar nature of the tricuspid annulus in showing the difference in tenting heights between orthogonal views: how does non-planarity affect the novel measurements?

Thank you for your clarification. This issue could be better addressed after testing all novel parameters in different study populations (including TR < 3+), comparing them by different methods, and predicting outcomes after TTVI. We have not control group/method/procedure to make definitive conclusions. Therefore, the study limitation section has been expanded to include these items.

  1. Although the main commissures AS, PS and AP are always identifiable on these 22 patients, there is clearly both pathologic, surgical and echocardiographic evidence for the additional complexity of the valve which significant impacts outcomes and cannot be underestimated. See: 1) Sugiura A, Tanaka T, Kavsur R, Öztürk C, Vogelhuber J, Wilde N, Becher MU, Zimmer S, Nickenig G, Weber M. Leaflet Configuration and Residual Tricuspid Regurgitation After Transcatheter Edge-to-Edge Tricuspid Repair. JACC Cardiovasc Interv. 2021 Oct 25;14(20):2260-2270. doi: 10.1016/j.jcin.2021.07.048. Epub 2021 Aug 23. PMID: 34424200, and 2) Kitamura M, Kresoja KP, Besler C, Leontyev S, Kiefer P, Rommel KP, Otto W, Forner AF, Ender J, Holzhey DM, Abdel-Wahab M, Thiele H, Borger MA, Hahn RT, Lurz P, Noack T. Impact of Tricuspid Valve Morphology on Clinical Outcomes After Transcatheter Edge-to-Edge Repair. JACC Cardiovasc Interv. 2021 Jul 26;14(14):1616-1618. doi: 10.1016/j.jcin.2021.03.052. Epub 2021 May 26. PMID: 34052157. Your current discussion does not accurately reflect the importance of these complex morphologies.

Thank you for this essential comment. This is exclusively an anatomical study of patients suffering from severe TR and no data of procedural success are available (Please see limitation section). However, your comment gave us some exciting inputs that we discussed.

  1. The study by van Rosendael (van Rosendael PJ, Joyce E, Katsanos S, Debonnaire P, Kamperidis V, van der Kley F, Schalij MJ, Bax JJ, Ajmone Marsan N, Delgado V. Tricuspid valve remodelling in functional tricuspid regurgitation: multidetector row computed tomography insights. Eur Heart J Cardiovasc Imaging. 2016 Jan;17(1):96-105. doi: 10.1093/ehjci/jev140. Epub 2015 Jun 9. PMID: 26060205) showed no difference in eccentricity index with severe vs non-severe TR however the AP dimension, was highly predictive of TR severity. Were you able to correlate any of your measures with TR severity?

Thank you very much for your suggestion, but we currently have no CT-scan of patients with less than severe TR, in order to apply our protocol. The goal is to make this comparison to give more strength to our analysis.

Reviewer 2 Report

This article overall reads well. The authors sought to identify CT parameters for the tricuspid valve CT analysis. Although this manuscript will give readers a better understanding of tricuspid valve anatomy in the presence of severe tricuspid regurgitation patients populations, I have a few remarks for the authors.

  1. The patients' sample size of 22 is too small to conclude.
  2. I would ask the authors to explain the methodology better and include an illustration that shows all the steps of the methodology.

Author Response

This article overall reads well. The authors sought to identify CT parameters for the tricuspid valve CT analysis. Although this manuscript will give readers a better understanding of tricuspid valve anatomy in the presence of severe tricuspid regurgitation patients populations, I have a few remarks for the authors.

  1. The patients' sample size of 22 is too small to conclude.

Thank you very much for your comment. Regrettably, we are aware of this strong limitation highlighted in the study limitations section. However, there are no other studies that have analyzed these parameters, and the goal of our research is to extend this analysis to patients with less than severe TR, to compare the CT scan parameters with parameters derived from other imaging modalities, and to evaluate the role of these variables in the context of TTVI procedures. For this reason, we reported that this study can be considered preliminary and hypothesis-generating.

  1. I would ask the authors to explain the methodology better and include an illustration that shows all the steps of the methodology.

 Thank you for your essential comment that allowed us to expand the methods section. According to your suggestion, we added an illustration that shows all the methodology steps (Figure S1). However Figure 1 presents a diagram of the novel CT scan measurements, while Figure 2 summarizes the CT scan views for analyzing tethering (degree and coaptation depth) and AROA